# Trends in social exposure to SARS-Cov-2 in France. Evidence from the national socio-epidemiological cohort–EPICOV

Josiane Warszawski[1,2]*, Laurence Meyer[1,2], Jeanna-Eve Franck[3], Delphine Rahib[4], Nathalie Lydié[4], Anne Gosselin[5,6], Emilie Counil[5], Robin Kreling[1], Sophie Novelli[1], Remy Slama[7,8], Philippe Raynaud[9], Guillaume Bagein[9], Vianney Costemalle[9], Patrick Sillard[10], Toscane Fourie[11], Xavier de Lamballerie[11], Nathalie Bajos[3,12], Epicov Team[¶]

1 INSERM CESP U1018, Université Paris-Saclay, Le Kremlin-Bicêtre, France, 2 AP-HP Epidemiology and Public Health Service, Service, Hôpitaux Universitaires Paris-Saclay, Le Kremlin-Bicêtre, France, 3 Iris–Institut de Recherche Interdisciplinaire sur les enjeux sociaux, Inserm, Aubervilliers, France, 4 Santé Publique France, Saint-Maurice, France, 5 French Institute for Demographic Studies (INED), Aubervilliers, France, 6 French Collaborative Institute on Migrations/CNRS, Aubervilliers, France, 7 Institut thématique de Santé Publique, INSERM, Paris, France, 8 Inserm, CNRS, Team of Environmental Epidemiology applied to Reproduction and Respiratory Health, Institute for Advanced Biosciences, University Grenoble Alpes, Grenoble, France, 9 DREES—Direction de la Recherche, des Etudes, de l'évaluation et des statistiques, Paris, France, 10 Institut National de la statistique et des études économiques, Montrouge, France, 11 Unité des Virus Emergents, UVE, Aix Marseille Univ, INSERM 1207, IRD 190, Marseille, France, 12 Ecole des Hautes Etudes en Sciences Sociales, Paris, France

¶ Membership of the Epicov Team author group can be found in the Acknowledgments.
* josiane.warszawski@inserm.fr

**Data Availability Statement:** All anonymous aggregated data concerning the results presented in this paper are available online and on supporting information files. The EpiCov study is available for

## Abstract

### Background

We aimed to study whether social patterns of exposure to SARS-CoV-2 infection changed in France throughout the year 2020, in light to the easing of social contact restrictions.

### Methods

A population-based cohort of individuals aged 15 years or over was randomly selected from the national tax register to collect socio-economic data, migration history, and living conditions in May and November 2020. Home self-sampling on dried blood was proposed to a 10% random subsample in May and to all in November. A positive anti-SARS-CoV-2 ELISA IgG result against the virus spike protein (ELISA-S) was the primary outcome. The design, including sampling and post-stratification weights, was taken into account in univariate and multivariate analyses.

### Results

Of the 134,391 participants in May, 107,759 completed the second questionnaire in November, and respectively 12,114 and 63,524 were tested. The national ELISA-S seroprevalence was 4.5% [95%CI: 4.0%-5.1%] in May and 6.2% [5.9%-6.6%] in November. It increased markedly in 18-24-year-old population from 4.8% to 10.0%, and among second-generation immigrants

research purpose after submission to approval of French Ethics and Regulatory Committee procedure (Comité du Secret Statistique, CESREES and CNIL). Access procedure is available on CASD (https://www.casd.eu/). Additional information can be addressed to the corresponding author.

**Funding:** This research was supported by research grants from Inserm (Institut National de la Santé et de la Recherche Médicale) and the French Ministry for Research, by Drees-Direction de la Recherche, des Etudes, de l'Evaluation et des Statistiques, and the French Ministry for Health, and by the Région Ile de France. Dr. Bajos has received funding from the European Research Council (ERC) under the European Union's Horizon 2020 research and innovation programme (grant agreement No. [856478]) This project has also received funding from the European Union's Horizon 2020 research and innovation programme under grant agreement No 101016167, ORCHESTRA (Connecting European Cohorts to Increase Common and Effective Response to SARS-CoV-2 Pandemic). The funders had no role in study design, data collection and analysis, decision to publish, or preparation of the manuscript.

**Competing interests:** The authors have declared that no competing interests exist

from outside Europe from 5.9% to 14.4%. This group remained strongly associated with sero-positivity in November, after controlling for any contextual or individual variables, with an adjusted OR of 2.1 [1.7–2.7], compared to the majority population. In both periods, seroprevalence remained higher in healthcare professions than in other occupations.

## Conclusion

The risk of Covid-19 infection increased among young people and second-generation migrants between the first and second epidemic waves, in a context of less strict social restrictions, which seems to have reinforced territorialized socialization among peers.

## Introduction

Social determinants contribute to socioeconomic, ethno-racial and spatial inequalities in COVID-19 exposure and severity [1, 2]. Their role may change over time according to the stringency or duration of social contact restrictions [3] and vaccination policies. African, Asian, Latin-American and other ethnic minorities were disproportionately affected by SARS-CoV-2 in Europe and North America during the first epidemic wave [4–8]. However, in the UK, the difference in age-standardized COVID-19 mortality between people with black ethnic background and the white population decreased markedly between the first and second waves [9].

France has been severely affected by COVID-19. The first wave peaked two weeks after the first national lockdown initiated on 17th March 2020, in a context of mask shortages and little availability of PCR tests. The first lockdown, which ended on 11th May 2020, after a dramatic decrease to a very low incidence rate, was very strict, with closure of schools, universities, cultural and social venues, shops except for essential supply, teleworking, and limitation of outdoor circulation.

The second wave started slowly at the end of August, despite a wide-scale distribution of masks and free access to PCR and antigenic tests. Following a period of mandatory physical-distancing and curfew with territorial variations, a second national lockdown was instated from 30 October to 15 December 2020. Unlike the first lockdown which caused widespread suspension of both social and professional life, the second was less restrictive, with no school closure and extended list of shops authorized to remain open. Between the first and second lockdown, teleworking was encouraged, measures maintaining barriers to extra-professional social life remained, especially face covering and maximum numbers admitted to access attractions, coffees and restaurant, but which let more opportunities to get together, especially during the summer.

Most analysis of social and ethnic disparities are based on mortality, hospitalization, and virologic PCR data. Here, we aimed to study the social dynamics of the epidemic between the end of the first lockdown in May and the second in November 2020, using the French national EpiCoV cohort, a large random population-based seroprevalence study [10], enabling identification of changes in factors associated with seropositivity in the context of the easing of social contact restrictions.

## Materials and methods

### Study design

Individuals aged 15 years or older living in France were randomly selected from the FIDELI administrative sampling frame, covering 96.4% of the population, providing postal addresses for all, and e-mail addresses or telephone numbers for 83%. FIDELI is the national database

on housing and individuals issued from tax files, containing demographic information on people and household structure and income, and additional contextual data about the living place of people. The sampling design is detailed elsewhere [10]. Differential sampling was used to ensure oversampling of the less densely populated *départements* (i.e French administrative districts), and lower-income categories. Residents in nursing homes for elderly persons were excluded, as it was not feasible to obtain help from caregivers to facilitate telephone or web contact with them during the first lockdown. All selected individuals were contacted by post, e-mail and text messages, with up to seven reminders. In the first round in May, computer-assisted-web interviews (CAWI) or computer-assisted-telephone interviews (CATI) were offered to a random 20% subsample. The remaining 80% were assigned to CAWI exclusively. All first-round respondents were eligible for the second in November 2020.

## Home capillary blood self-sampling for serological testing

This was proposed during the web/telephone questionnaire to a national random subsample in May, and to all respondents in November. Dried-blood spots were collected on 903Whatman paper (DBS) kits sent to each participant agreeing to blood sampling, mailed to three biobanks (Bordeaux, Amiens, Montpellier) to be punched with a PantheraTM machine (Perkin Elmer). Eluates were processed in a virology laboratory (Unité des virus Emergents, Marseille) with commercial ELISA kits (Euroimmun®, Lübeck, Germany) to detect anti-SARS-CoV-2 antibodies (IgG) against the S1 domain of the viral spike protein (ELISA-S), according to the manufacturer's instructions.

## Outcome

SARS-Cov-2 seroprevalence was estimated as the proportion of individuals tested with an ELISA-S ratio $\geq 1.1$, according to the threshold specified by the manufacturer.

## Exposure

Contextual living conditions included administrative geographical area, population density in the municipality of residence, whether the neighbourhood was defined as socially deprived with prioritizing of socio-economic interventions, the number of people in the household, the household per capita income decile, and whether any other household member had had a positive virological PCR or Antigen test since January 2020. Individual characteristics included gender, age, personal and parental migration history, educational level, current occupation (collected with more detail in November), tobacco use, body mass index and comorbidities, number of contacts and face mask use outside home in the week before the second-round interview.

## Ethics and regulatory issues

The survey was approved by CNIL (the French data protection authority) (ref: MLD/MFI/AR205138) and the ethics committee (Comité de Protection des Personnes Sud M editerranee III 2020-A01191-38) on April 2020, and by the "Comité du Label de la Statistique Publique". The serological results were sent to the participants by post with information about interpreting individual test results.

## Statistical analyses

We first repeated the same univariate and multivariate analyses on the May and November samples to estimate, for each period, the seroprevalence on national level and by geographical

area, contextual variables, housing conditions, and individual characteristics, and to study changes in the strength of their associations with the presence of antibodies between these two periods. We then considered the subsample of people tested negative in May (ELISA-S ratio <0.7), to study associations with positive serology in November, as a measure of the incidence of new infections between the two periods. Finally, we performed an additional multivariate analysis on the November sample, as it was much larger and included more detailed information than in May, that we added step by step in order to study the role of socio-economic and migration status more fully.

Final calibrated weights were calculated to correct for non-response, as detailed elsewhere [10], for first and second round. Response homogeneity groups were derived from the sampling weight divided by the probability of response estimated with logit models adjusted for auxiliary variables potentially linked to both the response mechanism and the main variables of interest in the EpiCov survey. The Fideli sampling frame provided a wide range of auxiliary variables, including sociodemographics, income, quality of contact information, and contextual variables at territorial level, such as population density, proportion of people below the poverty line, obtained from geo-referenced information. Variables collected in the first round were added as auxiliary variables to adjust non-response models for the second round. First-step weights estimated from the percentage of respondents in each homogeneity group were calibrated according to the margins of the population census data and population projections for age categories, gender, *departement*, educational level, and region, to decrease the variance and the residual bias for variables correlated with margins.

The unequal probabilities sampling design, and final calibrated weights were taken into account, with the specific design-based "proc survey" procedures of SAS and "svy" procedures of STATA. Prevalences were estimated, using weighted percentages, and logit transformed confidence limits were used to remain within the interval [0,1]. The design-based Pearson chi-squared test statistic developed by Rao was used for multiway contingency tables [11]. Crude and adjusted odds ratios were estimated with logistic regression models based on design-based methods [12]. The significance threshold was 0.05.

## Results

Among the 134 391 respondents to the first-round questionnaire in May 2020, 107 759 (80.2%) completed the second-round questionnaire in November 2020 (Fig 1). Serological tests were performed in mainland France on 12 114 participants for the first round (median date: May 21st 2020; IQR: 18th– 28th May), and 63 524 for the second (November 24th 2020; IQR: 18th November– 4th December).

The national seroprevalence (ELISA-S ratio ≥1.1) increased from 4.5% [95%CI: 4.0–5.1%] in May to 6.2% [5.9–6.6%] in November, with wide disparities between *départements* from under 2% to 13% (Table 1; S1 Table).

In both periods, seroprevalence was significantly higher among individuals living in highly densely populated municipalities, in socially deprived neighbourhoods and in large households (Table 1). The strength of the association with household size was weaker in November than in May.

Seroprevalence, which tended to be higher among women than men in May (5.0% versus 3.9%; p = 0.054), was similar between men and women in November (6.1% and 6.3%; p = 0.52) (Table 2). Seroprevalence increased with higher diploma levels, and was associated with a U-shaped curve with family *per capita* income, with lowest rates in the central decile especially in May. Prevalence remained nearly twice as high among healthcare professionals as among people with other occupations, whether self-reported as essential or not, respectively

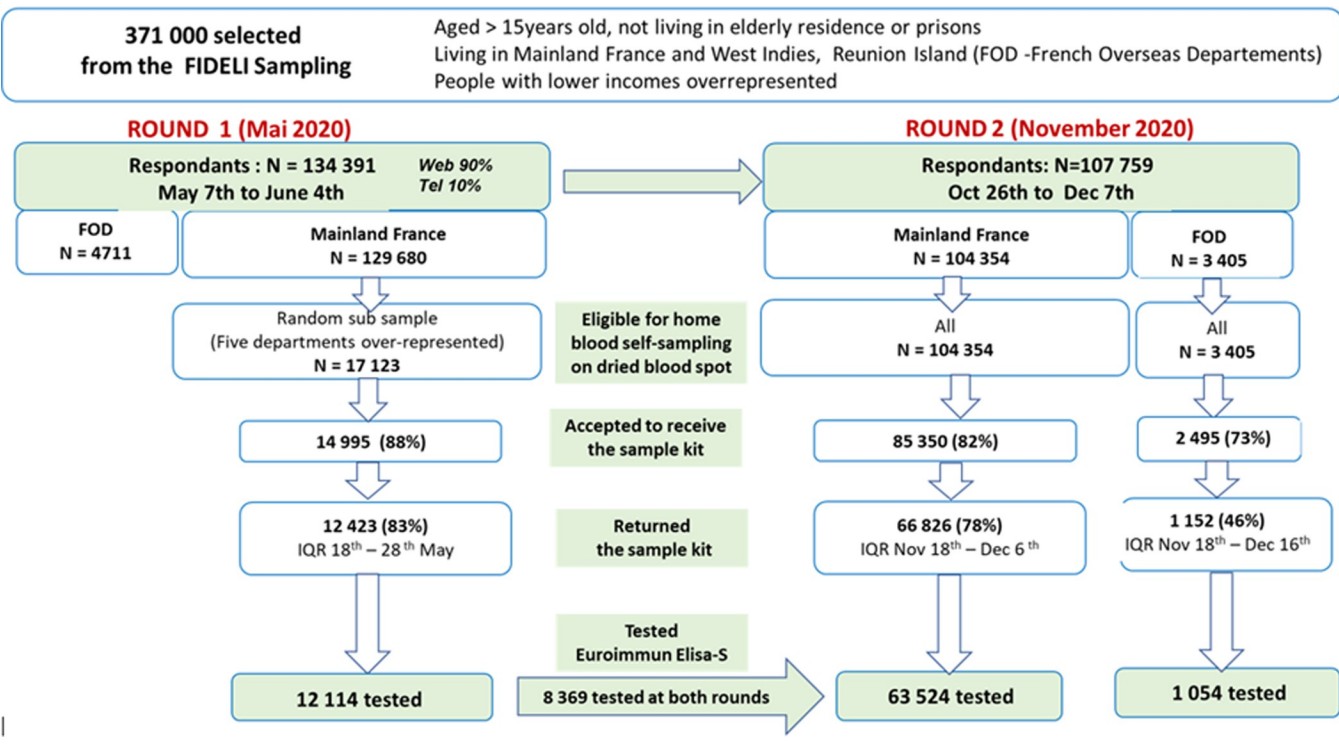

**Fig 1. Flowchart: The national EpiCov cohort, round 1 (May 2020) and round 2 (November 2020).**

11.3% and 6.4% in November. Detailed analysis of professional occupations in November showed the highest seroprevalences in hospital professions (physicians, nurses and assistant nurses), two to three times higher than for other occupations, including private physicians, pharmacists, teachers and workers in essential stores. Daily smokers were at lower risk of having antibodies than occasional, former or non-smokers.

The major changes in seroprevalence between May and November 2020 concerned age and migration status. In May 2020, the highest prevalence was observed among middle-aged people (8.3% in 35–44 years old) while in November 2020, it concerned the youngest (9.6% and 9.9% respectively in the 15–17 and 18–24 age groups). In May 2020, prevalence was significantly higher among immigrants born outside Europe (9.2% compared to 5.9% among second-generation immigrants from outside Europe, and 4.1% in the French-born population), but the increased risk disappeared after adjustment for living conditions (Table 3). In contrast, in November 2020, seroprevalence was higher in both first (13.3%) and second (14.4%) generation immigrants from outside Europe, compared to 5.3% among French-born and 6.0% among European immigrants, and they remained at higher risk even after adjustment for living conditions (adjusted odds ratio respectively: 2.1 [1.7–2.8] and 2.2 [1.8–2.9]).

In order to understand the overexposure of non-European immigrants and their descendants in November 2020, detailed analyses were performed (S2 Table), taking into account behaviours related to social distancing strategies self-reported over the week before the interview (number of prolonged contacts, mask use in the street, family or festive outings) and BMI. Associations with migration status remained unchanged. The analysis was also restricted to highly densely populated areas, and the overexposure of the second generation immigrants from outside Europe remained.

**Table 1. SARS-Cov-2 SEROPREVALENCE (ELISA-S ≥ 1.1[1]) according to living condition, among people living in mainland France [2]: The national EpiCov cohort, rounds 1 & 2.**

| | ELISA ≥ 1.1 (May 2020) [3] | | | | | ELISA ≥ 1.1 (November 2020) [3] | | | | |
|---|---|---|---|---|---|---|---|---|---|---|
| | Total | cases | % | CI 95% | p | Total | cases | % | CI95% | P |
| **All** | **12114** | **785** | **4.5** | **[4.0–5.1]** | | **63524** | **3943** | **6.2** | **[5.9–6.6]** | |
| Number of people in household | | | | | | | | | | |
| 1 | 1665 | 74 | 2.1 | [1.4–3.1] | <0.001 | 10377 | 570 | 5.2 | [4.5–5.9] | <0.001 |
| 2 | 4266 | 203 | 2.7 | [2.2–3.4] | | 24994 | 1331 | 4.9 | [4.6–5.3] | |
| 3 | 2268 | 173 | 5.1 | [4.0–6.6] | | 10902 | 741 | 6.5 | [5.8–7.2] | |
| 4 | 2560 | 210 | 7.1 | [5.6–8.9] | | 12040 | 899 | 7.9 | [7.1–8.8] | |
| 5 or more | 1349 | 125 | 8.5 | [6.1–11.8] | | 5189 | 400 | 10.1 | [8.7–11.8] | |
| ≥1 suspected Covid case in household | | | | | <0.001 | | | | | |
| Living alone | 1665 | 74 | 2.1 | [1.4–3.1] | | 10377 | 570 | 5.2 | [4.5–5.9] | <0.001 |
| No (and not living alone) | 8822 | 433 | 4.0 | [3.4–4.7] | | 37355 | 1494 | 4.1 | [3.8–4.5] | |
| Yes before June 2020 | 1621 | 278 | 12.9 | [10.6–15.6] | | 4543 | 514 | 12.0 | [10.4–13.8] | |
| Yes since June 2020 | | | | | | 8143 | 966 | 13.4 | [12.2–14.6] | |
| Yes before and after June 2020 | | | | | | 3084 | 397 | 12.6 | [11.1–14.3] | |
| Population density in municipality | | | | | | | | | | |
| Low | 3666 | 219 | 3.4 | [2.7–4.4] | <0.001 | 23647 | 1178 | 4.5 | [4.1–4.8] | <0.001 |
| Medium | 3562 | 199 | 3.3 | [2.5–4.2] | | 18650 | 1075 | 5.4 | [4.9–6] | |
| High | 4886 | 367 | 6.4 | [5.3–7.6] | | 21227 | 1690 | 8.5 | [7.9–9.2] | |
| Socially-deprived neighbourhood | | | | | | | | | | |
| No | 11589 | 743 | 4.2 | [3.7–4.8] | 0.021 | 61840 | 3778 | 5.9 | [5.6–6.2] | <0.001 |
| Yes | 525 | 42 | 8.2 | [4.7–14] | | 1684 | 165 | 11.2 | [8.9–14] | |
| Geographical area (region) | | | | | | | | | | |
| 11- Ile de France | 2430 | 214 | 9.0 | [7.3–11.2] | <0.001 | 10441 | 1021 | 11.0 | [10;0–12.1] | <0.001 |
| 24-Centre Loire | 232 | 8 | 2.4 | [1.2–5.0] | | 2527 | 107 | 4.2 | [3.1–5.7] | |
| 27-Bourgogne Franche Comté | 280 | 7 | 1.5 | [0.6–3.4] | | 3056 | 195 | 5.6 | [4.6–6.7] | |
| 28-Normandie | 266 | 7 | 1.5 | [0.7–3.3] | | 2788 | 115 | 3.1 | [2.5–3.8] | |
| 32-Hauts de France | 1499 | 66 | 3.7 | [2.2–6.1] | | 5876 | 418 | 6.8 | [5.9–7.9] | |
| 44-Grand Est | 3239 | 323 | 6.7 | [5.2–8.5] | | 6461 | 501 | 6.7 | [5.9–7.6] | |
| 52-Pays de Loire | 328 | 11 | 2.9 | [1.6–5.3] | | 3869 | 148 | 3.0 | [2.4–3.8] | |
| 53-Bretagne | 307 | 12 | 4.8 | [2.3–9.8] | | 3510 | 105 | 2.5 | [1.9–3.2] | |
| 75-Nouvelle Aquitaine | 538 | 13 | 2.0 | [1.1–3.5] | | 5820 | 202 | 3.4 | [2.8–4.1] | |
| 76-Occitanie | 560 | 19 | 2.2 | [1.4–3.7] | | 6335 | 268 | 4.5 | [3.7–5.5] | |
| 84-Auvergne | 716 | 36 | 4.0 | [2.8–5.6] | | 8274 | 643 | 8.4 | [7.4–9.4] | |
| 93-PACA | 1687 | 69 | 5.0 | [3.2–7.6] | | 4278 | 211 | 4.4 | [3.5–5.4] | |
| 94-Corse | 32 | 0 | 0.0 | | | 289 | 9 | 4.8 | [1.9–11.4] | |

1. Home sampling by finger prick/Euroimmun ELISA-S test

2. People aged 15 years or over residing in mainland France, outside nursing homes for elderly and prisons.

3. The sampling design is taken into account for the estimation of prevalence, confidence intervals (logit transformation) and statistical tests, with the SAS procsurvey procedure. The percentages are weighted by sampling weight (the inverse of inclusion probability), corrected for non-response weigts and calibrated on the margin of the census. The prevalences are not equal to n/N.

Results from the analysis of incidence of new infections between May and November was consistent with changes in seroprevalence (S3 Table). Overall, 3.8% [3.1–4.7%] of 7 515 people with no IgG antibodies in May were positive in November. The proportion of new infections was the highest in the 18–24 age group, among second-generation immigrants from outside Europe, among people living in socially deprived neighbourhoods, and among health-care

**Table 2. SARS-Cov-2 SEROPREVALENCE (ELISA-S ≥ 1.1[1]) according to individual socio-economic factors, among people living in mainland France [2]: The national EpiCov cohort, rounds 1 & 2.**

| | ELISA ≥ 1.1 (May 2020) [3] | | | | | ELISA ≥ 1.1 (November 2020) [3] | | | | |
|---|---|---|---|---|---|---|---|---|---|---|
| | Total | cases | % | CI 95% | p | Total | cases | % | CI95% | P |
| **All** | **12114** | **785** | **4.5** | **[4.0–5.1]** | | **63524** | **3943** | **6.2** | **[5.9–6.6]** | |
| Gender | | | | | | | | | | |
| Men | 5469 | 321 | 3.9 | [3.1–4.8] | 0.052 | 27564 | 1665 | 6.1 | [5.7–6.6] | 0.459 |
| Women | 6645 | 464 | 5.0 | [4.3–5.9] | | 35960 | 2278 | 6.4 | [6–6.8] | |
| Age (years) | | | | | | | | | | |
| 15–17 | 418 | 27 | 4.5 | [2.2–8.9] | <0.001 | 1438 | 128 | 9.8 | [7.8–12.2] | <0.001 |
| 18–24 | 1042 | 61 | 4.8 | [3–7.6] | | 4919 | 483 | 10.0 | [8.6–11.5] | |
| 25–34 | 1544 | 118 | 5.0 | [3.7–6.7] | | 6816 | 490 | 7.2 | [6.3–8.3] | |
| 35–44 | 2050 | 198 | 8.3 | [6.7–10.4] | | 10345 | 671 | 6.5 | [5.8–7.4] | |
| 45–54 | 2340 | 176 | 4.9 | [3.9–6.2] | | 12596 | 850 | 6.5 | [5.9–7.2] | |
| 55–64 | 2234 | 122 | 4.8 | [3.3–7.1] | | 12879 | 710 | 5.3 | [4.8–5.8] | |
| 65–74 | 1727 | 64 | 1.8 | [1.2–2.7] | | 10611 | 462 | 4.3 | [3.8–4.9] | |
| 75+ | 759 | 19 | 0.7 | [0.4–1.4] | | 3920 | 149 | 3.7 | [2.9–4.7] | |
| Migratory status [4] | | | | | | | | | | |
| No (majority population) | 9769 | 612 | 4.1 | [3.5–4.7] | <0.001 | 54296 | 3172 | 5.3 | [5.1–5.6] | <0.001 |
| Immigrant from Europe | | | | | | | | | | |
| First- generation | 345 | 22 | 3.8 | [2–6.9] | | 1577 | 84 | 5.2 | [3.9–6.8] | |
| Second- generation | 668 | 39 | 3.8 | [2.4–5.9] | | 3164 | 197 | 6.0 | [4.9–7.3] | |
| Immigrant from outside Europe | | | | | | | | | | |
| First- generation | 606 | 61 | 9.2 | [6.2–13.6] | | 1760 | 207 | 13.3 | [10.7–16.3] | |
| Second- generation | 581 | 44 | 5.9 | [3.8–9.2] | | 1894 | 233 | 14.4 | [11.9–17.4] | |
| Detailed Migratory status [4] | | | | | | | | | | |
| No (majority population) | | | | | | | | | | |
| Born in Mainland France | 9646 | 596 | 4.0 | 3.4–4.6 | | 53697 | 3109 | 5.3 | [5.0–5.5] | |
| Born in FOD [5] | 56 | 8 | 12.4 | 5.3–26.3 | | 301 | 32 | 7.3 | [4.8–11.2] | |
| Parents born in FOD[5] | 67 | 8 | 3.3 | 1.2–9.2 | | 298 | 31 | 7.5 | [4.7–11.6] | |
| Immigrant from Europe | | | | | | | | | | |
| First- generation | 345 | 22 | 3.8 | [2–6.9] | | 1577 | 84 | 5.2 | [3.9–6.8] | |
| Second- generation | 668 | 39 | 3.8 | [2.4–5.9] | | 3164 | 197 | 6.0 | [4.9–7.3] | |
| 1st generation from outside Europe | | | | | | | | | | |
| Born in Africa | 356 | 35 | 7.4 | 4.5–12.1 | | 950 | 126 | 15.5 | [11.9–20.0] | |
| Born in Asia or elsewhere | 250 | 26 | 13.4 | 7.1–23.7 | | 810 | 81 | 9.4 | [6.8–12.8] | |
| 2nd generation from outside Europe | 581 | 44 | 5.9 | [3.8–9.2] | | 1894 | 233 | 14.4 | [11.9–17.4] | |
| Born in Africa | 385 | 29 | 6.8 | 4.0–11.4 | | 1181 | 156 | 15.6 | [12.3–19.6] | |
| Born in Asia or elsewhere | 196 | 15 | 4.1 | 1.9–8.4 | | 713 | 77 | 12.1 | [8.7–16.5] | |
| Occupation in May [6] | | | | | | | | | | |
| Healthcare profession | 578 | 74 | 11.4 | [8.2–15.6] | <0.001 | 3219 | 338 | 11.3 | [9.8–13] | <0.001 |
| Other essential profession | 1219 | 99 | 5.2 | [3.8–7.1] | | 6259 | 381 | 6.4 | [5.3–7.7] | |
| Non-essential profession | 4960 | 365 | 5.7 | [4.8–6.8] | | 24984 | 1619 | 6.4 | [5.9–6.9] | |
| No occupation | 5356 | 247 | 3.0 | [2.3–3.9] | | 29046 | 1605 | 5.7 | [5.3–6.2] | |
| Educational level | | | | | | | | | | |
| < High school diploma | 1908 | 98 | 3.2 | [2.2–4.8] | <0.001 | 8496 | 488 | 5.6 | [4.9–6.3] | <0.001 |
| High school diploma | 3922 | 204 | 3.3 | [2.6–4.2] | | 20384 | 1171 | 5.7 | [5.2–6.3] | |
| Secondary first degree diploma | 2435 | 184 | 6.4 | [5.0–8.0] | | 13509 | 835 | 6.9 | [6.3–7.6] | |
| ≥ Bachelor's degree | 3849 | 299 | 6.2 | [5.2–7.5] | | 21135 | 1449 | 7.3 | [6.8–7.9] | |

*(Continued)*

**Table 2.** (Continued)

| | ELISA ≥ 1.1 (May 2020) [3] | | | | | ELISA ≥ 1.1 (November 2020) [3] | | | | |
|---|---|---|---|---|---|---|---|---|---|---|
| | Total | cases | % | CI 95% | p | Total | cases | % | CI95% | P |
| **All** | **12114** | **785** | **4.5** | **[4.0–5.1]** | | **63524** | **3943** | **6.2** | **[5.9–6.6]** | |
| Family income per capita (deciles) | | | | | | | | | | |
| D01 (lowest) | 798 | 52 | 5.7 | [3.2–9.9] | 0.016 | 3672 | 241 | 8.2 | [6.7–10] | <0.001 |
| D02-D03 | 1430 | 86 | 4.8 | [3.5–6.7] | | 6481 | 385 | 6.2 | [5.3–7.3] | |
| D04-D05 | 1718 | 97 | 3.3 | [2.4–4.5] | | 9098 | 523 | 5.3 | [4.7–6.0] | |
| D06-D07 | 2423 | 128 | 2.9 | [2.2–3.8] | | 13252 | 785 | 5.9 | [5.4–6.5] | |
| D08-D09 | 3332 | 237 | 5.5 | [4.5–6.7] | | 18724 | 1147 | 6.1 | [5.7–6.6] | |
| D10 | 2112 | 159 | 6.0 | [4.7–7.6] | | 10880 | 766 | 7.0 | [6.5–7.6] | |
| Tobacco use | | | | | | | | | | <0.001 |
| Daily smoker | 1995 | 69 | 2.8 | [2.0–4.0] | 0.032 | 8949 | 266 | 2.7 | [2.3–3.2] | |
| Occasional smoker | 470 | 33 | 5.1 | [3.1–8.2] | | 2941 | 196 | 7.8 | [6.2–9.7] | |
| Ex smoker since epidemic | | | | | | 879 | 48 | 5.4 | [3.3–8.6] | |
| Ex-smoker before epidemic | 3888 | 253 | 4.5 | [3.5–5.8] | | 15895 | 940 | 5.7 | [5.2–6.3] | |
| Non-smoker | 5756 | 430 | 5.1 | [4.3–6.0] | | 34819 | 2492 | 7.5 | [7.1–8] | |

1. Home sampling by finger prick/Euroimmun ELISA-S test

2. People aged 15 years or over residing in mainland France, outside nursing homes for elderly and prisons.

3. The sampling design is taken into account for the estimation of prevalence, confidence intervals (logit transformation) and statistical tests, with the SAS procsurvey procedure. The percentages are weighted by sampling weight (the inverse of inclusion probability), corrected for non-response weigts and calibrated on the margin of the census. The prevalences are not equal to n/N.

4. Migratory status: Majority population = persons born in France who are neither first nor second-generation immigrants / First-generation immigrants: born non-French outside France and living permanently in France (including those who subsequently acquired French nationality) / Second-generation immigrants: born and living in France, with at least one parent a first-generation immigrant

5. FOD: French overseas *départements*

6. Self-reported in round 1: a) Healthcare professions Include medical and paramedical professionals, firefighters, pharmacists and ambulance drivers (but not including hospital cleaners, for example),.; b) Other essential professions included: Home helps or housekeepers, food shop workers, delivery drivers, public transportation drivers, cab drivers, bank customer services or reception staff, petrol station employees, police officers, postal workers, cleaning staff, security guards, construction workers, truck drivers, farmers and social workers), also self-reported.

professionals. Neither household size, diploma nor family income were associated with new infections between May and November.

## Discussion

Seroprevalence in France increased slowly from the end of the first lockdown to the second, from 4.5% [95%CI: 4.0–5.1%] in May 2020 to 6.2% [5.9–6.6%] in November 2020. Seroprevalence estimated in November probably underestimates the cumulate incidence from the start of the epidemic, as the level of antibodies wanes with time [13–15]. However only 8.3% [7.3–9.4] of participants tested twice were positive at least once, and the highest prevalence rates were under 20% even in the most affected regions. At the end of 2020, the level of herd immunity in the general population in France remained low. Wide geographical disparities, with continental eastern and central areas the most affected, and western oceanic areas the least, could partly reflect the residual impact of the first strict national lockdown which stopped the spread of the virus from the north-east [16].

Between May and November 2020 seroprevalence increased much more among young people, while the middle-aged population was mainly affected during the first wave. This change is likely to be explained by more infections during the summer holidays and autumn,

**Table 3. Univariate and multivariate logistic regressions: Factors associated with ELISA-S seropositivity[1] among people living in mainland France at the end of first and second lockdown [2]: The national EpiCov cohort, rounds 1 & 2.**

| | ELISA ≥ 1.1 (May 2020) | | | | ELISA ≥ 1.1 (November 2020) | | | |
|---|---|---|---|---|---|---|---|---|
| | $OR_{crude}$ | 95% CI[3] | $OR_{adj}$ | 95% CI[3] | $OR_{crude}$ | 95% CI[3] | $OR_{adj}$ | 95% CI[3] |
| **Individual characteristics** | | | | | | | | |
| Gender | P = 0.053 | | P = 0.085 | | P = 0.45 | | P = 0.88 | |
| Men | ref | | ref | | ref | | ref | |
| Women | 1.3 | [1.0–1.7] | 1.3 | [1.0–1.7] | 1.0 | [0.9–1.2] | 1.0 | [0.9–1.1] |
| Age (years) | P<0.001 | | P = 0.003 | | P<0.001 | | P<0.001 | |
| 15–17 | 6.3 | [2.3–17.1] | 3.2 | [1.0–10.3] | 2.8 | [2.0–4.1] | 2.0 | [1.4–3.0] |
| 18–24 | 6.9 | [3.0–15.6] | 2.9 | [1.2–6.0] | 2.9 | [2.2–3.9] | 2.2 | [1.6–3.0] |
| 25–34 | 7.2 | [3.4–14.9] | 2.5 | [1.5–8.2] | 2.0 | [1.5–2.7] | 1.6 | [1.1–2.3] |
| 35–44 | 12.3 | [6.1–25.0] | 5.4 | [2.4–12.5] | 1.8 | [1.4–2.4] | 1.3 | [0.9–1.8] |
| 45–54 | 7.0 | [3.4–14.2] | 3.6 | [1.6–8.4] | 1.8 | [1.4–2.4] | 1.5 | [1.1–2.1] |
| 55–64 | 6.9 | [3.1–15.1] | 4.8 | [2.0–11.6] | 1.5 | [1.1–1.9] | 1.4 | [1.0–1.9] |
| 65–74 | 2.5 | [1.1–5.5] | 2.3 | [1.0–5.2] | 1.2 | [0.9–1.5] | 1.2 | [0.9–1.6] |
| 75+ | ref | | ref | | ref | | ref | |
| Migration status [4] | P = 0.002 | | P = 0.705 | | P<0.001 | | P<0.001 | |
| No (majority population) | ref | | ref | | ref | | ref | |
| First- generation from Europe | 0.9 | [0.5–1.8] | 1.1 | [0.6–2.3] | 1.0 | [0.7–1.3] | 1.1 | [0.8–1.4] |
| Second- generation Europe | 0.9 | [0.6–1.5] | 1.0 | [0.6–1.7] | 1.1 | [0.9–1.4] | 1.2 | [1.0–1.5] |
| First-generation outside Europe | 2.4 | [1.5–3.9] | 1.6 | [0.8–3.0] | 2.7 | [2.1–3.5] | 2.0 | [1.5–2.5] |
| Second- generation outside Europe | 1.5 | [0.9–2.5] | 1.1 | [0.6–1.9] | 3.0 | [2.4–3.8] | 2.1 | [1.7–2.7] |
| Occupational status [5] | P<0.001 | | P = 0.002 | | P<0.001 | | P = 0.001 | |
| Healthcare profession | 2.1 | [1.4–3.2] | 2.1 | [1.4–3.2] | 1.9 | [1.6–2.2] | 1.9 | [1.6–2.3] |
| Other essential profession | 0.9 | [0.6–1.3] | 1.0 | [0.7–1.5] | 1.0 | [0.8–1.2] | 0.9 | [0.8–1.1] |
| Non-essential profession | ref | | ref | | ref | | ref | |
| No occupation | 0.5 | [0.4–0.7] | 0.8 | [0.6–1.2] | 0.9 | [0.8–1.0] | 1.0 | [0.9–1.1] |
| Educational level | P<0.001 | | P = 0.072 | | P<0.001 | | P = 0.31 | |
| < High school diploma | ref | | ref | | ref | | ref | |
| High school diploma | 1.0 | [0.6–1.7] | 1.0 | [0.6–1.6] | 1.0 | [0.9–1.2] | 1.1 | [0.9–1.3] |
| Secondary first degree diploma | 2.0 | [1.3–3.3] | 1.5 | [0.9–2.5] | 1.3 | [1.1–1.5] | 1.2 | [1.0–1.5] |
| ≥ Bachelor's degree | 2.0 | [1.3–3.1] | 1.2 | [0.7–1.9] | 1.3 | [1.1–1.6] | 1.1 | [0.9–1.4] |
| Family income per capita (deciles) | P<0.001 | | P = 0.004 | | P<0.001 | | P = 0.009 | |
| D01 (lowest) | 2.0 | [1.0–3.9] | 1.5 | [0.8–2.9] | 1.4 | [1.1–1.8] | 1.1 | [0.8–1.3] |
| D02-D03 | 1.7 | [1.1–2.6] | 1.7 | [1.0–2.6] | 1.1 | [0.9–1.3] | 0.9 | [0.7–1.1] |
| D04-D05 | 1.1 | [0.7–1.7] | 1.1 | [0.7–1.7] | 0.9 | [0.8–1.0] | 0.8 | [0.7–1.0] |
| D06-D07 | ref | | ref | | ref | | ref | |
| D08-D09 | 1.9 | [1.4–2.7] | 1.9 | [1.3–2.7] | 1.0 | [0.9–1.2] | 1.0 | [0.9–1.2] |
| D10 | 2.1 | [1.5–3.1] | 1.9 | [1.3–2.9] | 1.2 | [1.1–1.4] | 1.2 | [1.0–1.3] |
| Tobacco use | P = 0.035 | | P = 0.025 | | P<0.001 | | P<0.001 | |
| Daily smoker | ref | | ref | | ref | | ref | |
| Occasional smoker | 1.8 | [1.0–3.5] | 2.0 | [1.0–4.0] | 3.1 | [2.3–4.2] | 2.4 | [1.7–3.3] |
| Ex smoker since epidemic | | | | | 2.1 | [1.2–3.5] | 1.9 | [1.1–3.2] |
| Ex-smoker before epidemic | 1.6 | [1.0–2.6] | 1.8 | [1.2–2.9] | 2.2 | [1.8–2.7] | 2.4 | [2.0–3.0] |
| Non-smoker | 1.8 | [1.2–2.8] | 1.9 | [1.2–2.8] | 3.0 | [2.5–3.6] | 2.8 | [2.3–3.5] |
| **Living conditions** | | | | | | | | |
| Population density in municipality | P<0.001 | | P <0.001 | | P<0.001 | | P<0.001 | |
| Low | ref | | ref | | ref | | ref | |

*(Continued)*

**Table 3.** (Continued)

| | ELISA ≥ 1.1 (May 2020) | | | | ELISA ≥ 1.1 (November 2020) | | | |
|---|---|---|---|---|---|---|---|---|
| | OR$_{crude}$ | 95% CI[3] | OR$_{adj}$ | 95% CI[3] | OR$_{crude}$ | 95% CI[3] | OR$_{adj}$ | 95% CI[3] |
| Medium | 1.0 | [0.7–1.4] | 1.1 | [0.8–1.6] | 1.2 | [1.1–1.4] | 1.1 | [1.0–1.3] |
| High | 1.9 | [1.4–2.7] | 1.8 | [1.3–2.5] | 2.0 | [1.8–2.2] | 1.6 | [1.4–1.8] |
| Socially deprived neighbourhood | P = 0.024 | | P = 0.35 | | <0.001 | | P = 0.009 | |
| No | ref | | ref | | ref | | ref | |
| Yes | 2.0 | [1.1–3.7] | 1.4 | [0.7–2.6] | 2.0 | [1.5–2.6] | 1.4 | [1.1–1.9] |
| Number of people in household | P<0.001 | | P = 0.003 | | P<0.001 | | P<0.001 | |
| 1 | ref | | ref | | ref | | ref | |
| 2 | 1.3 | [0.8–2.1] | 1.4 | [0.9–2.3] | 1.0 | [0.8–1.1] | 1.0 | [0.8–1.2] |
| 3 | 2.5 | [1.6–4.1] | 2.0 | [1.2–3.6] | 1.3 | [1.1–1.5] | 1.1 | [1.0–1.4] |
| 4 | 3.6 | [2.2–5.8] | 2.5 | [1.4–4.4] | 1.6 | [1.3–1.9] | 1.4 | [1.1–1.6] |
| 5 or more | 4.4 | [2.5–7.6] | 3.4 | [1.7–6.6] | 2.1 | [1.7–2.6] | 1.4 | [1.1–1.7] |

1. Home sampling by finger prick/Euroimmun ELISA-S test

2. People aged 15 years or over residing in mainland France, outside nursing homes and prisons.

3. The sampling design is taken into account for the estimation of prevalence, confidence intervals (logit transformation), crude and adjusted odds ratios, confidence intervals and statistical tests, with the SAS procsurvey procedure. The percentages are weighted by sampling weight (the inverse of inclusion probability), corrected for non-response weigts and calibrated on the margin of the census. The prevalences are not equal to n/N.

4. Migratory status: Majority population = persons born in France who are neither first nor second-generation immigrants / First-generation immigrants: born non-French outside France and living permanently in France (including those who subsequently acquired French nationality) / Second-generation immigrants: born and living in France, with at least one parent being a first-generation immigrant

5. Self-Reported in round 1: a) Healthcare professions Included medical and paramedical professionals, firefighters, pharmacists and ambulance drivers (but not including hospital cleaners, for example); b) Other essential professions included: home helps or housekeepers, food shop workers, delivery drivers, public transportation drivers, cab drivers, bank customer service or reception staff, petrol station employees, police officers, postal workers, cleaning staff, security guards, construction workers, truck drivers, farmers and social workers), also self-reported.

consistent with the higher positivity rate on PCR and antigenic tests reported to French Si-Dep surveillance systems between June and November 2020, ranging from 4.7% among 20-29-year-olds to 3.1% among 40-59-year-olds and 1.7% among 70-79-year-olds.

The second major change was the increased seroprevalence among descendants of non-European immigrants (second-generation immigrants), independently of their younger age. In May 2020, seroprevalence was twice as high among first-generation immigrants from outside Europe as in the majority population, i.e. neither immigrants nor their descendants, and this was mainly explained by residence in a densely-populated area and a large household. In November 2020, prevalence was three times higher among both non-European immigrants and their descendants, reflecting a strong increase in new infections in the second generation between May and November. Adjustment for age accounted for only part of this increase. Mostly, the association remained independent of socio-economic and living conditions, geographical area, mask use and number of prolonged contacts. Nor was this explained by differences in tobacco use, comorbidities or BMI. Similar results were observed when the analysis was restricted to highly-densely populated municipalities, and urban areas where most immigrants reside, or to areas the most affected by Covid-19.

African Americans, Hispanics, and other ethnic minority groups were disproportionately affected by SARS-CoV-2, as mostly documented during the first epidemic wave in terms of diagnosed infection, hospitalization [6–8] and mortality [7, 8]. Among potential reasons for higher incidence or severity related to ethnicity, biological susceptibilities have been hypothesized [17–19] but without evidence [20]. Inequalities in mortality could be primarily driven by

differences in exposure to infection [21]. In England, there was a marked reduction in the difference in age-standardized COVID-19 mortality between people from black ethnic backgrounds and people from the white group between first and second wave [9]. Some minority ethnic populations have excess risks of testing positive for SARS-CoV-2 and of adverse COVID-19 outcomes compared with the white population, even after taking account of differences in socio-demographic, clinical, and household characteristics [22].

Our study, based on repeated general population seroprevalence measures, showed that while the overexposure to Covid-19 infection of first-generation immigrants was strongly linked to their living conditions at the beginning of the epidemic, the overexposure observed six months later for the first and especially the second generation, who have more social contacts more than their elders, is not the result of a lesser respect for barrier gestures or of more frequent outings than the native-born (S1 Table). It could result from micro-social structural effects, because of the phenomena of socio-spatial segregation [23] and territorialized socialization [24]. Second-generation immigrants are very often grouped together, facilitating the circulation of the virus in social groups where the prevalence is higher.

Relationships between seropositivity and population density in the residence area, family income and diploma tended to be weaker in November than in May. This could suggest a protective role of the widespread use of masks in working and public areas, and testing strategies before visiting family. In a national survey in the UK, having patient-facing role and working outside home was an important risk factor in the first but not the second wave [25]. However, despite wide availability of surgical masks after severe shortage during the first epidemic wave, seroprevalence among healthcare professionals remained twice as high as among individuals with other occupations in November, similar to May, with the highest rates among hospital physicians, nurses and assistant nurses. The seroprevalence was similar in May and November while the proportion of new infections was much higher than in other occupations, which could suggest that health-care professionals were infected early during the first wave, with possibly higher proportions of seroreversion in that population because IgG levels decrease with time. This increased risk was not explained by socio-demographic or living conditions, except for medical students where the association was partly explained by their younger age. The 11% seroprevalence found in May is in line with the 8.5% found in Europe during the first wave in a meta-analysis [26], with few studies on the risk of nosocomial transmission among health-care worker [27].

## Strengths

The EpiCov cohort is one of the largest socio-epidemiological random population-based cohorts providing Covid-19 seroprevalence estimate among individuals aged 15 years and over. Most seroprevalence surveys were conducted during the first epidemic wave [28–30]. The two other national serological studies based on random general population samples was conducted in Spain [31] and England [21] and reported prevalence of same magnitude than in France. EpiCov identified the population most affected by the spread of the virus in the population since initial spread, providing a basis for evaluating subsequent changes in light with epidemiological context and access to preventive strategies. People living below the poverty line were intentionally over-represented in the sampling, and detailed socio-economic and migration data was obtained. We were therefore able to perform a powerful analysis focusing on social inequalities.

The home self-sampling with DBS detection of SARS CoV-2 antibodies was ideally suited to the context of the first lockdown to limit self-selection bias.

The estimates provided here were weighted for non-response. Many auxiliary demographic and socio-economic variables were available from the sampling framework, which made it possible to correct a large part of the non-response bias.

## Limitations

The EpiCov study had several limitations. It does not cover elderly people living in nursing homes. The Euroimmun ELISA-S test has a sensitivity of 94.4%, according to the manufacturer's cutoff. It has been evaluated in various studies, which reported specificity ranging from 96.2% to 100% and sensitivity ranging from 86.4% to 100% [32, 33]. Anti-Sars-Cov2 IgG antibody levels have been reported to decline more or less rapidly, particularly among the elderly and subjects with mild or asymptomatic forms [13–15]. However, such decline seems not to be a source of bias to study changes is population exposed to covid between the two epidemic waves: our results were similar when analysing factors associated with new Covid infections between May and November in the subsample tested in both rounds, and changes in factors associated with seroprevalence between these two periods.

## Conclusion

The role of living conditions on the risk of SARS-CoV-2 infection decreased between the first and second epidemic waves, possibly partly due to the widespread availability of masks and virological tests at population level. Nevertheless, in November 2020, in a context of less restricted social contacts than during the first lockdown, seroprevalence remained higher among healthcare professionals than among other professionals, and strongly increased among young people and racial minorities. These populations need special attention, especially for adherence to vaccination policies.

## Supporting information

**S1 Table. Seroprevalence (ELISA-S > 1.11) according to *département* in November 2020 among people living in mainland France.** The national EpiCov cohort, 2020 November round.
(DOCX)

**S2 Table. Factors associated with seropositivity (ELISA-S > 1.1) in November 2020 among people living in mainland France.** The national EpiCov cohort, 2020 November round–Univariate and multivariate analysis including detailed occupation, detailed living conditions and self-reported distancing behaviours over the last 7 days.
(DOCX)

**S3 Table. Proportion of new infections between May and November 2020: Proportion of positive serologies in November among people seronegative in May—The national EpiCov cohort.**
(DOCX)

## Acknowledgments

**Epicov Team**: Josiane Warszawski (co-principal investigator) and Nathalie Bajos (co-principal investigator), Guillaume Bagein, François Beck, Emilie Counil, Florence Jusot, Nathalie Lydie, Claude Martin, Laurence Meyer, Philippe Raynaud, Alexandra Rouquette, Ariane Pailhé, Delphine Rahib, Patrick Sicard, Rémy Slama, Alexis Spire.

Lead : Josiane Warszawski, INSERM CESP U1018, AP-HP Epidemiology and Public Health Service, S Université Paris-Saclay, Le Kremlin-Bicêtre, France Mail Josiane.warszawski@universite-paris-saclay.fr

Nathalie Bajos (co-lead), Iris–Institut de Recherche Interdisciplinaire sur les enjeux sociaux, Inserm, Aubervilliers, France ; Ecole des Hautes Etudes en Sciences Sociales, Paris, France

Guillaume Bagein, DREES—Direction de la Recherche, des Etudes, de l'évaluation et des statistiques, Paris, France

François Beck, Santé Publique France, Saint-Maurice France

Emilie Counil, French Institute for Demographic Studies (INED)

Florence Jusot, Université Paris Dauphine, Paris, France

Nathalie Lydie, Santé Publique France, Saint-Maurice France

Claude Martin, ARENES UMR 6051, CNRS, EHESP, Rennes, Franc

Laurence Meyer, INSERM CESP U1018, AP-HP Epidemiology and Public Health Service, S Université Paris-Saclay, Le Kremlin-Bicêtre, France Mail Josiane.warszawski@universite-paris-saclay.fr

Philippe Raynaud, DREES—Direction de la Recherche, des Etudes, de l'évaluation et des statistiques, Paris, France

Alexandra Rouquette, INSERM CESP U1018, AP-HP Epidemiology and Public Health Service, S Université Paris-Saclay, Le Kremlin-Bicêtre, France Mail Josiane.warszawski@universite-paris-saclay.fr

Ariane Pailhé,, French Institute for Demographic Studies (INED)

Delphine Rahib, Santé Publique France, Saint-Maurice France

Patrick Sillard, Institut National de la statistique et des études économiques, Montrouge, France

Rémy Slama, Institut thématique de Santé Publique, INSERM, Paris France, Inserm, CNRS, Team of Environmental Epidemiology applied to Reproduction and Respiratory Health, Institute for Advanced Biosciences, University Grenoble Alpes, Grenoble, France

Alexis Spire, Iris–Institut de Recherche Interdisciplinaire sur les enjeux sociaux, Inserm, Aubervilliers, France

We sincerely thank all the participants in the EpiCoV study.

We warmly thank the INSERM staff, including, in particular, Carmen Calandra, Karim Ammour, Jean-Marc Boivent, Jean-Marie Gagliolo, Frédérique Le Saulnier, and Frédéric Robergeau, who worked with considerable dedication and commitment to make it possible to develop, in record time, and to maintain all regulatory, budgetary, technical, and logistical aspects of the EpiCov study.

We warmly thank the staff of Santé publique France, and especially Lucie Duchesne, who played a major role in organisation and quality assurance for the seroprevalence component of the EpiCov study.

We thank the CRB biobanks staff, and especially their heads, Dr Isabelle Pellegrin, and Julien Jeanpetit (Centre Hospitalier Universitaire Robert Pellegrin, Bordeaux, France), Pr Edouard Tuaillon Centre de Ressources Biologiques du CHU de Montpellier), Dr Yves-Edouard Herpe (Centre de Ressources Biologiques Biobanque de Picardie), Pr Jacqueline Deloumeaux (Centre biologique du CHU de la Guadeloupe), Dr Rémi Neviere (CeRBiM, Centre de Ressources Biologiques de la Martinique), Julien Eperonnier, Estelle Nobecourt (Centre de Ressources Biologiques de la Réunion) for the quality of DBS sample management of the EpiCov study. We thank the biobank team in Inserm SC10, particularly Sophie Circosta.

We also thank the staff of the UVE virology department, for the high-quality management of such a large number of serological assays.

We thank the staff of DREES and INSEE, for their collaboration in the implementation of the study, methodological input, sample selection, and the complex development of weights to correct for non-response.

We thank the Ipsos staff, including Christophe David and Valérie Blineau in particular, for their major contribution to the quality of data collection.

## The EPICOV study group

Josiane Warszawski[1,2], Nathalie Bajos[9] (joint principal investigators), Guillaume Bagein, François Beck[4], Emilie Counil[10], Florence Jusot[11], Nathalie Lydié[4], Claude Martin[12], Laurence Meyer[,2], Philippe Raynaud[7], Alexandra Rouquette[1,2], Ariane Pailhé[10], Delphine Rahib[4], Patrick Sillard[8], Alexis Spire[12].

INSERM CESP U1018, Université Paris-Saclay, Le Kremlin-Bicêtre, France

[2] AP-HP Epidemiology and Public Health Service, Service, Hôpitaux Universitaires Paris-Saclay, Le Kremlin-Bicêtre, France

[3] Unité des Virus Emergents, UVE, Aix Marseille Univ, IRD 190, INSERM 1207, IHU Méditerranée Infection, Marseille, France

[4] Santé Publique France, Saint-Maurice France

[5] Institut thématique de Santé Publique, INSERM, Paris France

[6] Inserm, CNRS, Team of Environmental Epidemiology applied to Reproduction and Respiratory Health, Institute for Advanced Biosciences, University Grenoble Alpes, Grenoble, France

[7] DREES—Direction de la Recherche, des Etudes, de l'évaluation et des statistiques, Paris, France

[8] Institut National de la statistique et des études économiques, Montrouge, France

[9] IRIS, INSERM, EHESS, CNRS Aubervilliers, France

[10] INED, France

[11] Université Paris Dauphine, France

[12] CNRS, France

**The EPICOV study group**: J Warszawski, N Bajos (Co-PI), G Bagein, F Beck, E Counil, F Jusot,' N Lydié, C Martin, L Meyer, P Raynaud, A Rouquette, A Pailhé, D Rahib, P Sillard, A Spire.

## Author Contributions

**Conceptualization:** Josiane Warszawski, Laurence Meyer, Nathalie Lydié, Remy Slama, Philippe Raynaud, Patrick Sillard, Xavier de Lamballerie, Nathalie Bajos.

**Formal analysis:** Josiane Warszawski, Jeanna-Eve Franck, Robin Kreling, Sophie Novelli, Vianney Costemalle.

**Funding acquisition:** Remy Slama.

**Investigation:** Toscane Fourie, Xavier de Lamballerie.

**Methodology:** Josiane Warszawski, Laurence Meyer, Delphine Rahib, Philippe Raynaud, Patrick Sillard.

**Project administration:** Guillaume Bagein.

**Resources:** Delphine Rahib.

**Supervision:** Josiane Warszawski, Nathalie Bajos.

**Writing – original draft:** Josiane Warszawski, Laurence Meyer, Nathalie Bajos.

**Writing – review & editing:** Josiane Warszawski, Laurence Meyer, Jeanna-Eve Franck, Delphine Rahib, Nathalie Lydié, Anne Gosselin, Emilie Counil, Robin Kreling, Sophie Novelli, Remy Slama, Philippe Raynaud, Guillaume Bagein, Vianney Costemalle, Xavier de Lamballerie.

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
