## [Decision Letter · Decision Letter 0]

22 Dec 2021

PONE-D-21-33692Trends in social exposure to SARS-Cov-2 in France. Evidence from the national socio-epidemiological cohort – EPICOV Evidence from the national socio-epidemiological cohort – EPICOVPLOS ONE

Dear Dr. Warszawski,

Thank you for submitting your manuscript to PLOS ONE. After careful consideration, we feel that it has merit but does not fully meet PLOS ONE’s publication criteria as it currently stands. Therefore, we invite you to submit a revised version of the manuscript that addresses the points raised during the review process.

We look forward to receiving your revised manuscript.

Kind regards,

Dong Keon Yon, MD, FACAAI

Academic Editor

PLOS ONE

Journal Requirements:

"This research was supported by research grants from Inserm (Institut National de la Santé et de la Recherche Médicale) and the French Ministry for Research, by Drees-Direction de la Recherche, des Etudes, de l’Evaluation et des Statistiques, and  the French Ministry for Health,  and by the Région Ile de France.

Dr. Bajos has  received funding from the European Research Council (ERC) under the European Union’s Horizon 2020 research and innovation programme (grant agreement No. [856478])

This project has also received funding from the European Union’s Horizon 2020 research and innovation programme under grant agreement No 101016167, ORCHESTRA (Connecting European Cohorts to Increase Common and Effective Response to SARS-CoV-2 Pandemic)."

3. Thank you for stating the following in the Financial Disclosure Section of your manuscript: 

"This research was supported by research grants from Inserm (Institut National de la Santé et de la Recherche Médicale) and the French Ministry for Research, by Drees-Direction de la Recherche, des Etudes, de l’Evaluation et des Statistiques, and  the French Ministry for Health,  and by the Région Ile de France.

Dr. Bajos has  received funding from the European Research Council (ERC) under the European Union’s Horizon 2020 research and innovation programme (grant agreement No. [856478])

This project has also received funding from the European Union’s Horizon 2020 research and innovation programme under grant agreement No 101016167, ORCHESTRA (Connecting European Cohorts to Increase Common and Effective Response to SARS-CoV-2 Pandemic)."

"This research was supported by research grants from Inserm (Institut National de la Santé et de la Recherche Médicale) and the French Ministry for Research, by Drees-Direction de la Recherche, des Etudes, de l’Evaluation et des Statistiques, and  the French Ministry for Health,  and by the Région Ile de France.

Dr. Bajos has  received funding from the European Research Council (ERC) under the European Union’s Horizon 2020 research and innovation programme (grant agreement No. [856478])

This project has also received funding from the European Union’s Horizon 2020 research and innovation programme under grant agreement No 101016167, ORCHESTRA (Connecting European Cohorts to Increase Common and Effective Response to SARS-CoV-2 Pandemic)."

5. One of the noted authors is a group or consortium Epicov Team. In addition to naming the author group, please list the individual authors and affiliations within this group in the acknowledgments section of your manuscript. Please also indicate clearly a lead author for this group along with a contact email address.

7. Please upload a new copy of Figure 2 as the detail is not clear. Please follow the link for more information: https://blogs.plos.org/plos/2019/06/looking-good-tips-for-creating-your-plos-figures-graphics/" https://blogs.plos.org/plos/2019/06/looking-good-tips-for-creating-your-plos-figures-graphics/

8. We note that Figure 2 in your submission contain [map/satellite] images which may be copyrighted. All PLOS content is published under the Creative Commons Attribution License (CC BY 4.0), which means that the manuscript, images, and Supporting Information files will be freely available online, and any third party is permitted to access, download, copy, distribute, and use these materials in any way, even commercially, with proper attribution. For these reasons, we cannot publish previously copyrighted maps or satellite images created using proprietary data, such as Google software (Google Maps, Street View, and Earth). For more information, see our copyright guidelines: http://journals.plos.org/plosone/s/licenses-and-copyright.

Additional Editor Comments:

Please address the excellent comments from the reviewers.

Minor comments

1. please disucss the paper below.

Lee SW, Yuh WT, Yang JM, Cho YS, Yoo IK, Koh HY, Marshall D, Oh D, Ha EK, Han MY, Yon DK. Nationwide Results of COVID-19 Contact Tracing in South Korea: Individual Participant Data From an Epidemiological Survey. JMIR Med Inform. 2020 Aug 25;8(8):e20992. doi: 10.2196/20992. PMID: 32784189; PMCID: PMC7470235.

2. Title

Trends in social exposure to SARS-Cov-2 in France. Evidence from the national socioepidemiological cohort – EPICOV Evidence from the national socio-epidemiological cohort – EPICOV

->

Trends in social exposure to SARS-Cov-2: Results from the French Nationwide Cohort

Reviewers' comments:

Reviewer's Responses to Questions

**Comments to the Author**

1. Is the manuscript technically sound, and do the data support the conclusions?

Reviewer #1: Yes

Reviewer #2: Yes

2. Has the statistical analysis been performed appropriately and rigorously? 

Reviewer #1: Yes

Reviewer #2: No

3. Have the authors made all data underlying the findings in their manuscript fully available?

Reviewer #1: Yes

Reviewer #2: Yes

4. Is the manuscript presented in an intelligible fashion and written in standard English?

Reviewer #1: Yes

Reviewer #2: Yes

5. Review Comments to the Author

Reviewer #1: Trends in social exposure to SARS-Cov-2 in France. Evidence from the national socioepidemiological cohort – EPICOV Evidence from the national socio-epidemiological cohort – EPICOV

Title is adequate.

The abstract is well described and objective.

Keywords: I suggest to the authors to exclude some keywords as “random sample” and include keywords that remind the country the study was conducted.

Introduction: Very adequate.

“African, Asian and other ethnic minorities were disproportionately affected by SARS-CoV-2 in Europe and North America during the first epidemic wave.” Please include “Latin-American individuals”.

“France has been severely affected by COVID-19. The first wave peaked two weeks after the first lockdown initiated on 17th March” Please describe here the year. Also please describe in the text if it was a national lockdown or regional lockdown.

“The second wave started slowly at the end of August,” Same observation as described above regarding the year.

“but leaving more opportunities to get together from the summer.” Please clarify this statement .

Aim of the study is adequate and well-described.

Methods:

Please describe what is “FIDELI administrative sampling framework” ?

“Residents in nursing homes for elderly persons were excluded.” Please describe why.

Study design is very adequate.

In the Exposure section were evaluated medical conditions of the participants? It is not clear for me.

Results section is well-described. Tables are adequate too. I would suggest to exclude table 4 and describe its information in the text, due to a lot of tables in the study.

Discussion: The discussion section is well argued. I miss other data in the literature on serological surveys in France and in countries that have adopted the lockdown, such as England and Spain.

Strenghts and limitations are adequately described. Conclusion is adequate.

Reviewer #2: Thanks to extensive and well conducted epidemiological study in France, this work has revealed an increased risk of SARS-CoV-2 transmission in young people and second-generation migrants when restrictions were less stringent between the first two pandemic waves. However, the statistic analysis has not been well detailed in the manuscript. The authors talk about univariate and multivariate analyses, but a much more extensive explanation should be provided. Which statistical tests have been applied, how, and why? Do the data fulfill all requirements to apply these tests? What about the statistical potency? I guess it is high due to the very high sample number. Please, show all details. Please, present the data in an APA format or similar, associating the statistica value to the p-values that appear in tables. I guess the t-test has been applied because the confidence intervals are given, but statistical analysis is of utmost importance here and should be explained very rigorously. I guess that it has been properly performed, but an idea only exist as far as it is written, as Jacques Monod highlighted. The a priori chosen level of statistical significance should be indicated. The same is applicable to correlation and logistic regression analysis and inference.

As the authors point out in the "Limitations" section, circulating antibody titles may vary and decay over time, and even they disappear in certain cases. However, memory B cell analysis is not feasible in this context. In lines 385 through 387, the authors highlight consistency between factors associated to incidence and prevalence. However, this does not solve the limitation, and this should be also pointed out. In this sentence, "of new infections" should be removed, because this is included in the "incidence" concept.

Please, remove an extra space after period in line 391. There are also some extra spaces to remove in the supplementary file. Please, carefully review this.

6. PLOS authors have the option to publish the peer review history of their article (what does this mean?). If published, this will include your full peer review and any attached files.

Reviewer #1: **Yes: **Vicente Sperb Antonello

Reviewer #2: No

---

## [Author Response · Author response to Decision Letter 0]

25 Mar 2022

Editor comments

Q2 and 3 Financial disclosure :

Role of the funders: 

Response: The funders had no role in study design, data collection and analysis, decision to publish, or preparation of the manuscript 

Place in the manuscript: 

Response: the contains is strictly similar to those provided in the online Funding Statement and we removed it from the manuscript, and we will add the above sentence about the role of the funders 

Added in the cover letter

This research was supported by research grants from Inserm (Institut National de la Santé et de la Recherche Médicale) and the French Ministry for Research, by Drees-Direction de la Recherche, des Etudes, de l’Evaluation et des Statistiques, and the French Ministry for Health, and by the Région Ile de France.

Dr. Bajos has received funding from the European Research Council (ERC) under the European Union’s Horizon 2020 research and innovation programme (grant agreement No. [856478])

This project has also received funding from the European Union’s Horizon 2020 research and innovation programme under grant agreement No 101016167, ORCHESTRA (Connecting European Cohorts to Increase Common and Effective Response to SARS-CoV-2 Pandemic).

Q4 Title : Please amend either the title on the online submission form (via Edit Submission) or the title in the manuscript so that they are identical.

Response: keep title in the manuscript

Trends in social exposure to SARS-Cov-2 in France. Evidence from the national socio-epidemiological cohort – EPICOV

Q5 Group epicov : One of the noted authors is a group or consortium Epicov Team. In addition to naming the author group, please list the individual authors and affiliations within this group in the acknowledgments section of your manuscript. Please also indicate clearly a lead author for this group along with a contact email address

Response: Done

Q6 “data not shown” in the manuscript : 

Response: Such data are not necessary for the current paper and they are developed in a submitted paper currently in revision elsewhere. We then suppressed the sentence which referred to it as it is not necessary. Our discussion and conclusion are entirely supported by data included in the paper (line 336 to 341).

We suppressed the unnecessary sentence in the discussion “Populations of non-European first and second-generation immigrants were as compliant with barrier measures as others in March and November (data not shown)”. 

Q7 and 8 Please upload a new copy of Figure 2 as the detail is not clear. We note that Figure 2 in your submission contain [map/satellite] images which may be copyrighted. 

Response: It was too difficult to obtain total copyright. This figure is not needed as the Table 1 and the supplementary table (as referred) bring geographical seroprevalence. 

We suppressed the figure 2 

Additional Editor Comments:

Minor comments

1. please disucss the paper below.

Lee SW, Yuh WT, Yang JM, Cho YS, Yoo IK, Koh HY, Marshall D, Oh D, Ha EK, Han MY, Yon DK. Nationwide Results of COVID-19 Contact Tracing in South Korea: Individual Participant Data From an Epidemiological Survey. JMIR Med Inform. 2020 Aug 25;8(8):e20992. doi: 10.2196/20992. PMID: 32784189; PMCID: PMC7470235.

Response : please, can you explain what is expected as discussion? 

2.Title

Trends in social exposure to SARS-Cov-2 in France. Evidence from the national socioepidemiological cohort – EPICOV Evidence from the national socio-epidemiological cohort – EPICOV->

Trends in social exposure to SARS-Cov-2: Results from the French Nationwide Cohort

Response: we retain “Trends in social exposure to SARS-Cov-2 in France. Evidence from the national socio-epidemiological cohort – EPICOV”

 

Reviewer #1

 I thank the reviewer for the comments and enclose here our responses to proposals or questions.

Com1 : Keywords: I suggest to the authors to exclude some keywords as “random sample” and include keywords that remind the country the study was conducted.

Response: We think that the specificity of such design is very important to mention in keywords and propose to replace random sample by “probability sample design”. The country is mentioned in the title.

Keyword : random sample replaced by “probability sampling design”

Com2 : “African, Asian and other ethnic minorities were disproportionately affected by SARS-CoV-2 in Europe and North America during the first epidemic wave.” Please include “Latin-American individuals”

Response: Done

Added L62: African, Asian, Latin-American and other ethnic minorities were disproportionately affected by SARS-CoV-2 in Europe and North America during the first epidemic wave

Com3 : “France has been severely affected by COVID-19. The first wave peaked two weeks after the first lockdown initiated on 17th March” Please describe here the year. Also please describe in the text if it was a national lockdown or regional lockdown. The second wave started slowly at the end of August,” Same observation as described above regarding the year.

Response: Done

The first national lockdown initiated on 17th March 2020 (line 68)

A second national lockdown was instated from 30 October to 15 December 2020 (line 76)

Com6 : “ but leaving more opportunities to get together from the summer.” Please clarify this statement 

Response: I agree that it is not very clear. the paragraph was clarified 

Change line 77-82 : Unlike the first lockdown which caused widespread suspension of both social and professional life, the second was less restrictive, with no school closure and extended list of shops authorized to remain open. Between the first and second lockdown, teleworking was encouraged, measures maintaining barriers to extra-professional social life remained, especially face covering and maximum numbers admitted to access attractions, coffees and restaurant, but which let more opportunities to get together, especially during the summer.

Com7 : Please describe what is “FIDELI administrative sampling framework” ?

Response: done 

102 to 105 : FIDELI is the national database on housing and individuals issued from tax files, containing demographic information on people and household structure and income, and additional contextual data about the living place of people.

Com7 : “Residents in nursing homes for elderly persons were excluded.” Please describe why.

Response: done 

Added in lines 107 to 108 : Residents in nursing homes for elderly persons were excluded, as it was not feasible to obtain help from caregivers to facilitate telephone or web contact with them during the first lockdown.

Com9: In the Exposure section were evaluated medical conditions of the participants? It is not clear for me.

Response: Self reported symptoms and comorbidities were collected in the questionnaire. For this analysis, we did not include data on symptoms. Adjustment for some comorbidities to study the relation of seropositivity with migration status was performed and presented in supplemental material. 

We added in line 136 : Individual characteristics included …, body max index and comorbidities…..

Com9: Results section is well-described. Tables are adequate too. I would suggest to exclude table 4 and describe its information in the text, due to a lot of tables in the study.

Response: done

The Table 4 was removed and added to supplemental data S3

Com11: Discussion: The discussion section is well argued. I miss other data in the literature on serological surveys in France and in countries that have adopted the lockdown, such as England and Spain.

Response: done

Added in line 424 et 425 (with two added references) : The two other national serological studies based on random general population samples was conducted in Spain (ref Pollan) and England (ref Ward) and reported prevalence of same magnitude than in France. 

 

Reviewer #2

I thank the reviewer for the comments and enclose here our responses to proposals or questions.

Com1 : The authors talk about univariate and multivariate analyses, but a much more extensive explanation should be provided. Which statistical tests have been applied, how, and why? Do the data fulfill all requirements to apply these tests? What about the statistical potency? I guess it is high due to the very high sample number. Please, show all details. Please, present the data in an APA format or similar, associating the statistical value to the p-values that appear in tables. I guess the t-test has been applied because the confidence intervals are given, but statistical analysis is of utmost importance here and should be explained very rigorously. I guess that it has been properly performed, but an idea only exist as far as it is written, as Jacques Monod highlighted. The a priori chosen level of statistical significance should be indicated. The same is applicable to correlation and logistic regression analysis and inference.

Response: 

I agree with the importance to add more details on the statistical methods used in this paper (that I reduce because of the limits of words).

The methodology is adapted to complex sample design, as standard procedures based upon classical SRS (simple random sample) and IID (Independent and identically distributed random variables) are generally not appropriate in such design. There is a large amount of methodological literature and I added two major classical references on the principle of the méthods in that domain, with large discussion on design-based or model-based approach: 

Skinner CJ, Holt D, Smith TMF. Analysis of complex surveys [Internet]. John Wiley & Sons; 1989 [cited 2022 Feb 12]. 328 p. Available from: https://eprints.soton.ac.uk/34690/

Rao JN, Scott AJ. On chi-squared tests for multiway contingency tables with cell proportions estimated from survey data. Ann Stat. 1984;46–60. 

We used here the design-based statistical methods, classically used in many population-based study, using procedure developed and validated in Stata (svy procedures), SAS (proc survey), or R (package survey) 

I am not used to add the value of the statistics value with the p-value, and which is rarely presented in epidemiologic papers. Statistical value is not interpretable here as it does not correspond to classical tests. Moreover tests and confidence intervals cannot be calculated from frequency presented in the tables as weighting is applied for point estimate, and design is taken into account for variance estimation. 

I rewrited the methodological paragraph with reference to the two papers, and hope it will be sufficient. 

Paragraph rewritten (line 171-177): The unequal probabilities sampling design, and final calibrated weights were taken into account, with the specific design-based “proc survey” procedures of SAS and “svy” procedures of STATA. Prevalences were estimated, using weighted percentages, and logit transformed confidence limits were used to remain within the interval [0,1]. The design-based Pearson chi-squared test statistic developed by Rao was used for multiway contingency tables (12). Crude and adjusted odds ratios were estimated with logistic regression models based on design-based methods (11). The significance threshold was 0.05.

Com2 : What about the statistical potency? I guess it is high due to the very high sample number.

Response: Sample size was initially calculated so as to ensure sufficient precision for the seroprevalence estimate, the goal being to obtain a 95% confidence interval of 2 points for a prevalence of 5% in administrative subdivisions of 600,000 inhabitants (department or metropolitan area). Moreover individuals living in a household below the poverty line were overrepresented to have sufficient powerful to study relation of exposure with social disadvantage (as indicated in line 107-108) 

Com3 : As the authors point out in the "Limitations" section, circulating antibody titles may vary and decay over time, and even they disappear in certain cases. However, memory B cell analysis is not feasible in this context. In lines 385 through 387, the authors highlight consistency between factors associated to incidence and prevalence. However, this does not solve the limitation, and this should be also pointed out. In this sentence, "of new infections" should be removed, because this is included in the "incidence" concept.

Response: We agree that the decline of antibodies should be a limitation to study trends in prevalence. However, our objective was to study whether there were changes in population exposed to Covid between first and second epidemic waves. As our conclusions are very similar when analyzing new infection between May and November and when comparing factors associated with prevalences at each period, we can conclude than decline in antibodies was not a source of bias for our main results. 

Paragraph changed in 412-16 : However, such decline seems not to be a source of bias to study changes is population exposed to covid between the two epidemic waves: our results were similar when analysing factors associated with new Covid infections between May and November in the subsample tested in both rounds, and changes in factors associated with seroprevalence between these two periods.

---

## [Decision Letter · Decision Letter 1]

14 Apr 2022

Trends in social exposure to SARS-Cov-2 in France. Evidence from the national socio-epidemiological cohort – EPICOV

PONE-D-21-33692R1

Dear Dr. Warszawski,

We’re pleased to inform you that your manuscript has been judged scientifically suitable for publication and will be formally accepted for publication once it meets all outstanding technical requirements.

Kind regards,

Dong Keon Yon, MD, FACAAI

Academic Editor

PLOS ONE

Additional Editor Comments (optional):

This is an excellent paper.

Reviewers' comments:

Reviewer's Responses to Questions

**Comments to the Author**

1. If the authors have adequately addressed your comments raised in a previous round of review and you feel that this manuscript is now acceptable for publication, you may indicate that here to bypass the “Comments to the Author” section, enter your conflict of interest statement in the “Confidential to Editor” section, and submit your "Accept" recommendation.

Reviewer #1: All comments have been addressed

Reviewer #2: All comments have been addressed

2. Is the manuscript technically sound, and do the data support the conclusions?

Reviewer #1: Yes

Reviewer #2: Yes

3. Has the statistical analysis been performed appropriately and rigorously? 

Reviewer #1: Yes

Reviewer #2: Yes

4. Have the authors made all data underlying the findings in their manuscript fully available?

Reviewer #1: Yes

Reviewer #2: Yes

5. Is the manuscript presented in an intelligible fashion and written in standard English?

Reviewer #1: Yes

Reviewer #2: Yes

6. Review Comments to the Author

Reviewer #1: I have carefully read the review carried out by the authors. All my questions were answered properly and adjustments were made. Therefore, I recommend this article for publication in Plos One.

Reviewer #2: In my opinion, the manuscript is now ready for publication because all isues raised have been adequately addressed and the manuscript improved accordingly.

7. PLOS authors have the option to publish the peer review history of their article (what does this mean?). If published, this will include your full peer review and any attached files.

Reviewer #1: **Yes: **Vicente Sperb Antonello

Reviewer #2: No

---

## [Editor Report · Acceptance letter]

28 Apr 2022

PONE-D-21-33692R1 

Trends in social exposure to SARS-Cov-2 in France. Evidence from the national socio-epidemiological cohort – EPICOV 

Dear Dr. Warszawski:

I'm pleased to inform you that your manuscript has been deemed suitable for publication in PLOS ONE. Congratulations! Your manuscript is now with our production department. 

Kind regards, 

on behalf of

Dr. Dong Keon Yon 

Academic Editor

PLOS ONE